# Motion Characteristics of a Clutch Actuator for Heavy-Duty Vehicles with Automated Mechanical Transmission

**Yunxia Li** [1,2,*] **and Zengcai Wang** [3]

1   School of Mechanical & Automotive Engineering, Qilu University of Technology (Shandong Academy of Sciences), Jinan 250353, China
2   Shandong Institute of Mechanical Design and Research, Jinan 250353, China
3   School of Mechanical Engineering, Shandong University, Jinan 250061, China; wangzc@sdu.edu.cn
*   Correspondence: liyunxia@qlu.edu.cn

**Abstract:** Clutch control has a great effect on the starting quality and shifting quality of heavy-duty vehicles with automated mechanical transmission (AMT). The motion characteristics of a clutch actuator for heavy-duty vehicles with AMT are studied in this paper to investigate the clutch control strategy further. The modeling principle of the automatic clutch system is analyzed, and a simulation analysis is given to prove its validity and rationality. Normalized velocity and velocity modulation percentage are proposed as evaluation parameters for the clutch actuator driven by pulse width modulation (PWM) signals. Based on an AMT test bench, the actuator motion characteristics are analyzed. Experimental results show that the range of normalized velocity and velocity modulation percentage are obtained for the clutch engagement and disengagement processes. By analyzing the experimental data, the engaging velocity and disengaging velocity of the actuator are estimated using the solenoid valves in combination. The research results provide a fundamental basis for precise controlling of the clutch and improving the smoothness of heave-duty vehicles.

**Keywords:** AMT; clutch actuator; PWM control; motion characteristics; heavy-duty vehicle



## 1. Introduction

AMT has been widely used in heavy-duty vehicles for more than two decades because it has good advantages of large torque, high transmission efficiency, high reliability, and low cost [1–3], whereas other automated transmissions such as automatic transmission (AT), continuously variable transmission (CVT), and double clutch transmission (DCT) cannot meet the demands of the heavy-duty vehicles. The AMT typically has a high transmission efficiency of more than 90%, and its efficiency loss is mainly caused by discrete gear ratios and torque interruption during shifting [4,5]. In contrast, a push-belt CVT has a lower efficiency of 75% due to torque losses including hydraulic pumping torque loss, belt friction loss, and belt slip loss [4]. In low torque conditions, the transmission efficiency is less than 60% caused by the wet clutch and torque converter in the AT and the DCT [6,7]. Therefore, it is very difficult for the AT, CVT, and DCT to provide large torque and high transmission efficiency to meet the demands of heavy-duty vehicles [8–11]. Currently, the input torque of the AMT can reach to 3400 N·m supplied by ZF Group, which can meet the requirements of large transmitted torque for heavy-duty vehicles.

Up to now, limited by mechanical manufacturing technologies, dry clutch has been used to transmit torque in heavy-duty vehicles with AMT. Clutch engagement control is always the key difficulty for heavy-duty vehicles with AMT during starting and shifting [12–14]. To improve the starting and shifting qualities of heavy-duty vehicles, many researchers studied the control methods of the clutch. Gao et al. designed a nonlinear shaft torque observer for trucks with AMT to improve the longitudinal dynamic performances [1]. Gao et al. proposed an improved optimal controller for the start-up of trucks and also pointed out the deficiencies of the control performance influenced by the clutch torque [2].

Wang et al. proposed a two-layer structure control strategy of an AMT clutch during hill start for heavy-duty vehicles, but the hydraulic actuator's motion characteristics were not studied [12]. Szimandl et al. studied a dynamic hybrid model of an electro-pneumatic clutch system, but the studying was limited to simulation [14]. Song et al. proposed a double-layer control strategy based on a gas-assisted hydraulic clutch actuator, but the actuator's motion characteristics were not given [15]. Pisaturo et al. studied thermal compensation control strategy in automated dry clutch engagement for commercial vehicles and proved the effectiveness in terms of simulation, but the experimental results were lacking [16]. Gao et al. studied a nonlinear control method of a direct-drive pump-controlled clutch actuator in consideration of the pump efficiency map, but the motion characteristics of the hydraulic actuator were not studied in detail [17]. Based on the above analysis, the studies of the dry clutch in AMT have been mainly focused on the optimal control strategies. The motion characteristics of the clutch actuator have been less studied, which will influence and determine the control performances of the clutch in AMT.

Generally, the clutch actuator contains a relatively complicated system, including pump, hydraulic (pneumatic) cylinder, and high-speed valves. The motion characteristics of the clutch actuator are affected by the high-speed valves and the cylinder. Additionally, the motion of the actuator has distinct nonlinear and lagging phenomenon. The control performances of some optimal control methods depend on the basic motion characteristics of the clutch actuator.

The aim of this paper is to study the motion characteristics of a clutch actuator for heavy-duty vehicles with AMT, providing engaging and disengaging characteristics of the clutch actuator.

This paper is organized as follows. In Section 2, the modeling and analyzing of an automatic pneumatic actuator system for heavy-duty vehicles are derived. In Section 3, the evaluation parameters of normalized velocity, engaging velocity modulation percentage, and disengaging velocity modulation percentage are proposed. In Section 4, simulation analysis is discussed to prove the validity of the clutch modeling and analyzing, containing a numerical simulation based on MATLAB and a dynamic simulation based on AMESim. Section 5 is experimental results of the clutch actuator motion characteristics based on an AMT test bench. Section 6 is brief descriptions of the actuator motion characteristics from the two perspectives of a simulation and a test. Finally, some conclusions are given in Section 7.

## 2. Clutch Modeling and Analyzing

An automatic clutch system is given to illustrate the clutch control principle. Based on the dynamic equations, the nonlinear motion characteristics and lag performances of the clutch actuator are analyzed.

### 2.1. Automatic Clutch System

A pull-type clutch, which is named for the moving direction of the release bearing such that the clutch disengaging motion can be achieved if the release bearing is pulled away from the clutch, is used in pneumatic AMT for heavy-duty vehicles. Figure 1 shows an automatic clutch system for heavy-duty AMT consisting of a dry clutch assembly, a release fork, and a pneumatic clutch actuator (PCA). The dry clutch assembly is comprised of a pressure plate, a clutch disk, a diaphragm spring, and a release bearing. The PCA contains a clutch power cylinder, two engaging solenoid valves (EV1 is a quick engaging solenoid valve [QESV], and EV2 is a slow engaging solenoid valve [SESV]), and two disengaging solenoid valves (DV1 is a quick disengaging solenoid valve [QDSV], and DV2 is a slow disengaging solenoid valve [SDSV]).

Compressed air with a certain pressure is transmitted into the clutch power cylinder chamber without the piston rod if DV1 and DV2 are powered on while EV1 and EV2 are powered off. Then, the piston rod extends out. Based on the action of the release fork, the release bearing pulls the small end of the diaphragm, and the pressure plate moves

away from the clutch disk, which is the clutch disengaging motion. The compressed air in the clutch power cylinder chamber without a piston rod is exhausted into the atmosphere if EV1 and EV2 are powered on while DV1 and DV2 are powered off. Then, the piston moves back if the pressure of the clutch power cylinder chamber is decreased. Based on the action of the release fork, the release bearing pushes the small end of the diaphragm, and the pressure plate moves toward the clutch disk, which is the clutch engaging motion. An external opening is designed to ensure that the cylinder chamber with a piston rod is always connected with the atmosphere. The gas is expelled during the clutch disengaging process, whereas it is filled during the clutch engaging process.

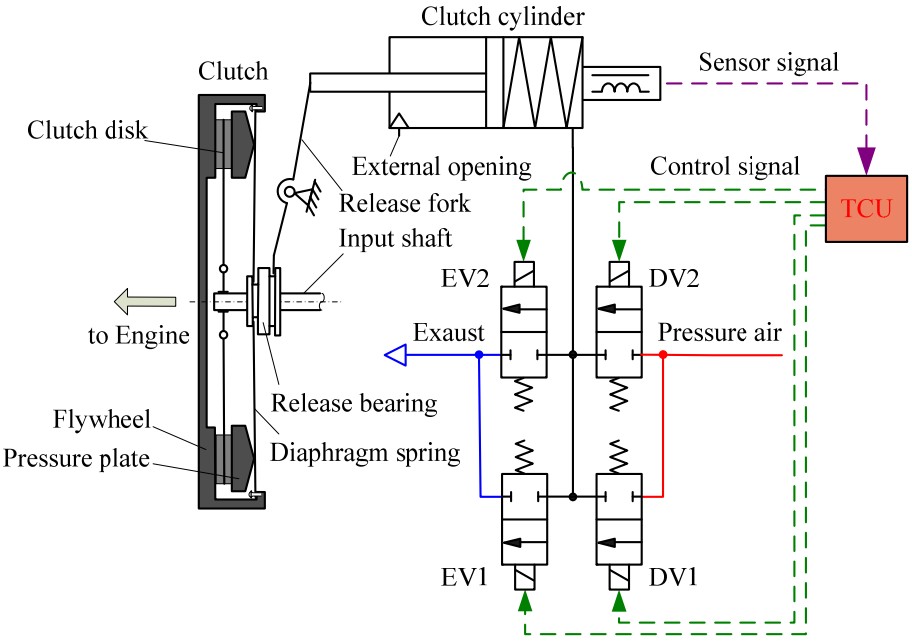

**Figure 1.** Schematic diagram of the automatic clutch system for heavy-duty AMT.

### 2.2. Solenoid Valve Modeling

The clutch solenoid valves are high-speed on–off valves, which are normally closed. The solenoid valve operation of the disengaging solenoid valve is shown in Figure 2. The schematic diagrams of the other solenoid valves are similar. The disengaging solenoid valve has two positions and two ways. When the solenoid valve is powered on in Figure 2b, the electromagnetic force generated from the solenoid overcomes the spring force that causes the spool move away from the seat until the valve is opened. When the solenoid valve is powered off in Figure 2a, the electromagnetic force is rapidly decreased to zero, which causes the spool to move back to the seat under the action of the spring force until the valve is closed.

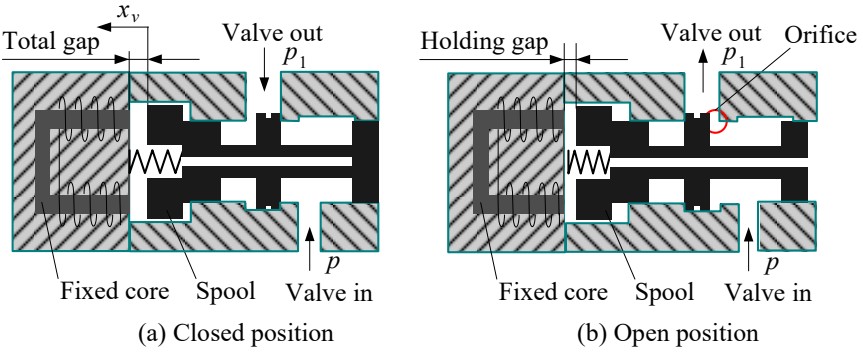

**Figure 2.** Schematic diagram of the solenoid valve operation.

The voltage equation of the solenoid is expressed in Equation (1).

$$U = (R_s + R_L)i + N\frac{d\Phi}{dt} \tag{1}$$

where $U$ is the driving voltage; $R_s$ is the inner resistance of source; $R_L$ is the resistance of the solenoid; $i$ is the current of the solenoid; $\Phi$ is the magnetic flux of every coil turn, and $N$ is the numbers of coil turns.

Ignoring the magnetic flux leakage, the main gap length is excited by the spool displacement. Then, the electromagnetic force of the spool can be expressed in Equation (2).

$$F_m = \frac{\mu_0 S_m N^2 i^2}{2(x_0 - x_v)^2} \tag{2}$$

where $F_m$ is the electromagnetic force of the spool; $\mu_0$ is the magnetic permeability; $S_m$ is the sectional area of the magnetic circuits, and $x_0$ and $x_v$ are the spool initial position (closed position) and the spool displacement, respectively.

The forces acting on the spool mainly include the force of the compressed air, the electromagnetic force, the spring force, the friction force, and the air resistance from the cylinder chamber without a piston rod. Therefore, the spool movement can be expressed in Equation (3).

$$\begin{cases} \frac{d^2 x_v}{dt^2} = \frac{1}{m_v}\left(F_a + F_m - F_s - F_f - F_p - C_v v_s\right) \\ \frac{dx_v}{dt} = v_s \end{cases} \tag{3}$$

where $F_a$ is the force of the compressed air expressed as $F_a = pA_v$; $p$ is the pressure of the pneumatic system; $A_v$ is the spool area; $F_s$ is the spring force expressed as $F_s = k_v(x_v + S_0)$; $k_v$ is the spring coefficient of the spool spring; $S_0$ is the pre-deformation of the spool spring; $F_f$ is the friction force of the spool moving; $F_p$ is the air resistance expressed as $F_p = p_1 A_o$; $p_1$ is the air pressure in the cylinder chamber without a piston rod; $A_o$ is the spool orifice area; $C_v$ is the damping coefficient of the spool moving; $m_v$ is the moving mass of the spool, and $v_s$ is the moving velocity of the spool.

It is shown from Equation (2) that the value of $F_m$ is related to the current of the solenoid and the spool displacement. The main gap length has its minimum value (holding gap) when the valve is fully open, and it has its maximum value (total gap) when the valve is fully closed. As a result, $F_m$ gets its maximum value if the valve is fully opened and gets its minimum value if the valve is fully closed.

The spool motion of the solenoid valve is mainly influenced by $F_m$, which is decided by $i$ and $x_v$, based on Equations (2) and (3). Therefore, the spool motion has stranger nonlinear characteristics, owing to the nonlinear relationships between $F_m$ and $i$ or $x_v$.

The spool cannot move if the electromagnetic force is not big enough after the solenoid is powered on, which can be shown from Equation (3). It needs a little time for the increasing of the electromagnetic force because the process of the electromagnetic field needs a little time after the solenoid is powered on. Similarly, it still needs a little time for the decreasing of the electromagnetic force after the solenoid is powered off. Therefore, the spool motion has some lag performances. Generally, the dynamic response time of opening or closing the solenoid valve is about 2 ms [18–20]. The dynamic response time decreases with the increasing of the voltage, the electric current, or the pressure [21–24].

Duty cycle is used to describe the percentage of the electric time for the solenoid valve. It can be expressed as Equation (4).

$$\alpha = \frac{t_h}{t_w} \times 100\% \tag{4}$$

where $\alpha$ is the duty cycle; $t_w$ is the cycle time of the solenoid valve, and $t_h$ is the time of high level that is less than or equal to $T$.

Controlling PWM signals can change the average flow of the solenoid valve because the opening degree of the solenoid valve is decided by the maximum of electromagnetic force limited by the average current. Based on high frequencies of PWM voltage signals, the solenoid valve can be considered to be in only two states: one state is in a condition of full opening, and the other is in a condition of full closing. Generally, the relationship between the flow of the solenoid valve and the duty cycle is approximately linear [25–29], which is shown in Figure 3.

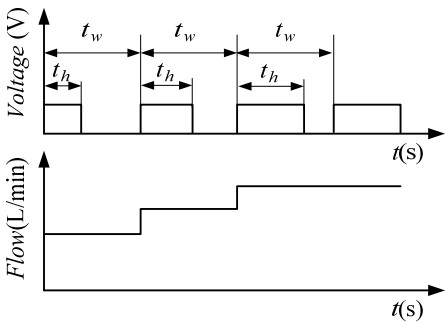

**Figure 3.** Relationship between the average flow and the duty cycle.

In this way, the relationship between the average flow and the duty cycle can be indicated as Equation (5).

$$Q = \alpha Q_{max} \tag{5}$$

where $Q$ is the average flow of the solenoid valve, and $Q_{max}$ is the maximum flow of the solenoid valve under the duty cycle of 100%.

### 2.3. Clutch Force and Transmitted Torque Analysis
#### 2.3.1. Clutch Force Analysis

A diagram of clutch force analyzing engaging and disengaging situations is shown in Figure 4. In situations of a clutch engaging, the piston of the clutch cylinder moves towards the cylinder chamber without a piston rod under the action of the release bearing and the release fork; meanwhile, the pressure plate and the friction disks are pressed together under the condition of the diaphragm. In situations of a clutch disengaging, the piston of the clutch cylinder moves towards the cylinder chamber without a piston rod under the action of the clutch cylinder because of filling with compressed air, and the pressure plate moves away from the clutch disks. In Figure 4, $x_l$ is the displacement of the diaphragm large end; $x_b$ is the displacement of the diaphragm small end (the displacement of the release bearing); $x_c$ is the piston displacement; $F_z$ is the force of the diaphragm acting on the pressure plate; $F_b$ is the force of the release bearing acting on the release fork, and $F_r$ is the force of the release fork acting on the piston rod.

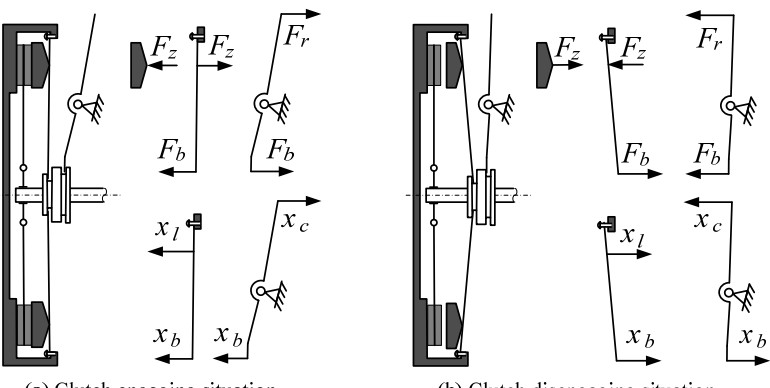

(a) Clutch engaging situation          (b) Clutch disengaging situation

**Figure 4.** Diagram of clutch force.

The Almen-Laszlo formula is often used to indicate the force of the diaphragm acting on the pressure plate [30–33]. The force of the diaphragm acting on the pressure plate can be expressed in Equation (6).

$$F_z = \frac{\pi E h x_l ln \frac{R}{r}}{6(1 - \mu_s)^2 (L - l)^2} \left\{ \left[ H - x_l \frac{R - r}{2(L - l)} \right] \left[ H - x_l \frac{R - r}{4(L - l)} \right] + h^2 \right\} \tag{6}$$

where $E$ is the modulus elasticity of the diaphragm; $\mu_s$ is Poisson's ratio; $h$ is the thickness of the diaphragm; $H$ is the height of the inner tapered section; $L$ is the radius of the rivet that fixes the diaphragm and the clutch cover together; $l$ is the radius of the force at the pressure plate; $R$ is the outside diameter of the diaphragm, and $r$ is the inside diameter of the diaphragm.

Considering the lever–ratio relationship, $k_d$ is defined as the lever ratio of the diaphragm from the small end to the large end, and $k_r$ is defined as the lever ratio of the release fork from the piston rod to the release bearing. The displacement of the diaphragm large end can be expressed as $x_l = x_b / k_d = x_c / (k_d k_r)$; the force of the release fork acting on the piston rod can be expressed as $F_r = F_b / k_r = F_z / (k_d k_r)$. Then, the relationship between the force of the piston rod and the displacement of the cylinder piston can be expressed in Equation (7) according to Equation (6).

$$F_r = \frac{\pi E h x_c ln \frac{R}{r}}{6 k_d^2 k_r^2 (1 - \mu_s)^2 (L - l)^2} \left\{ \left[ H - x_c \frac{R - r}{2 k_d k_r (L - l)} \right] \left[ H - x_c \frac{R - r}{4 k_d k_r (L - l)} \right] + h^2 \right\} \tag{7}$$

It shows that the force of the release fork acting on the piston rod has a stronger nonlinear relationship with the piston displacement from Equation (7). It is a cubic polynomial about the piston displacement of PCA.

### 2.3.2. Clutch-Transmitted Torque Analysis

The transmitted torque of the clutch is connected to the force of the pressure plate acting on the clutch disc, the coefficient ratio, and the efficient radius of the friction force acting on the clutch disk [30–33], which can be expressed in Equation (8) according to Equation (6).

$$T_c = \mu' Z F_Z R_e = \mu' Z \frac{\pi E h x_l ln \frac{R}{r}}{6(1 - \mu_s)^2 (L - l)^2} \left\{ \left[ H - x_l \frac{R - r}{2(L - l)} \right] \left[ H - x_l \frac{R - r}{4(L - l)} \right] + h^2 \right\} R_e \tag{8}$$

where $\mu'$ is the friction coefficient of the clutch disk; $Z$ is the number of the friction faces, and $R_e$ is the efficient radius of the friction force that can be expressed as $R_e = (D^3 - d^3) / 3(D^2 - d^2)$; $D$ is the outside diameter of the clutch disk, and $d$ is the inside diameter of the clutch disk.

Substituting $x_l = x_c / (k_d k_r)$ into Equation (8), the transmitted torque of the clutch can be indicated as a function of the piston displacement in Equation (9).

$$T_c = \mu' Z \frac{\pi E h x_c ln \frac{R}{r}}{6 k_d k_r (1 - \mu_s)^2 (L - l)^2} \left\{ \left[ H - x_c \frac{R - r}{2 k_d k_r (L - l)} \right] \left[ H - x_c \frac{R - r}{4 k_d k_r (L - l)} \right] + h^2 \right\} R_e \tag{9}$$

Equation (9) shows that the transmitted torque of the clutch is connected to the friction ratio of the clutch disc, the displacement of the cylinder piston, and the parameters of the diaphragm. The friction coefficient varies according to the clutch materials, the thermal deformation, the speed difference between the clutch driving plate and the clutch-driven plate [34–39]. As a result, the transmitted torque of the clutch is difficult to calculate accurately.

To sum up, the clutch-transmitted torque can be controlled by controlling the displacement of the clutch actuator (clutch cylinder). To reduce shock and vibration, the motion characteristics of the clutch actuator should be studied in depth for the purpose of improving starting and shifting qualities.

### 2.4. Clutch Cylinder Modeling

The diagram of the clutch power cylinder is shown in Figure 5. The forces during the engaging and disengaging situations are shown. A plus signal (+) denotes the engaging direction, and a minus signal (−) denotes the disengaging direction. Chamber 1 and chamber 2 are the cylinder chamber without a piston rod and one with a piston rod, respectively. The forces acting on the piston mainly include the force of the compressed air in the chamber 1, the spring force, the friction force, the force of the atmosphere in the chamber 2, and the force of the release fork acting on the piston rod.

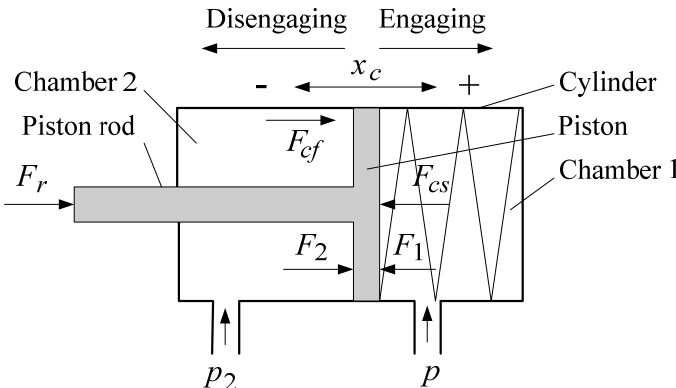

**Figure 5.** Diagram of the clutch power cylinder.

The piston motion equation can be expressed in Equation (10) based on the force condition of the cylinder above.

$$\begin{cases} \frac{d^2 x_c}{dt^2} = \frac{1}{m_c}\left( F_1 - F_2 - F_{cs} - F_{cf} - F_r - B_c v_c \right) \\ \frac{dx_c}{dt} = v_c \end{cases} \tag{10}$$

where $m_c$ is the mass of the piston and the piston rod; $v_c$ is the piston velocity; $F_1$ is the gas acting force on the piston in the cylinder chamber 1 that can be expressed as $F_1 = p A_1$; $p$ is the pressure of the pneumatic system; $A_1$ is the piston area; $F_2$ is the gas acting force on the piston in the cylinder chamber 2 that can be expressed as $F_2 = p_2 A_2$; $p_2$ is the atmospheric pressure in the cylinder chamber 2; $A_2$ is the gas acting area on the piston in the cylinder chamber 2 that equals the piston area minus the piston rod area; $F_{cs}$ is the cylinder spring force that can be expressed as $F_{cs} = k_c(x_c + G_0)$; $k_c$ is the elastic coefficient of the cylinder spring; $G_0$ is the pre-deformation of the cylinder spring; $F_{cf}$ is the friction force of the piston moving, and $B_c$ is the damping coefficient of the piston moving.

The force of the piston rod is related to the dynamic characteristics of the diaphragm and the lever ratio from the diaphragm's large end to the piston rod, which can be seen from Equation (7). The values of $F_1$, $F_2$, and $F_{cf}$ basically stay constant if the pressure of the pneumatic system is stable. The piston motion of PCA changes with $F_r$, which can be shown from Equation (10), and $F_r$ is influenced by the nonlinear characteristics of the diaphragm. Thus, the piston motion has stronger nonlinear characteristics decided by the nonlinear characteristics of the diaphragm.

Furthermore, the motion of the cylinder piston has some lag performances that can be seen from Equation (10). It takes little time to create the driving force enough to change the original condition. The lag time varies owing to the duty cycle, the atmospheric pressure, and the pressure of the pneumatic system.

The delay time from providing electric current to moving the piston comprises two aspects: one comes from the lag of the electric current of the solenoid, and the other one comes from the lag of the spool movement. It is difficult to control the clutch precisely for these lag performances of the actuator.

## 3. Main Evaluation Parameters of the Motion Characteristics

To compare the actuator moving velocity under a certain duty cycle with that under 100% duty cycle, normalized velocity is defined as an evaluation basis as expressed in Equation (11).

$$\overline{v} = \frac{v_\alpha}{v_{100}} \times 100\% \tag{11}$$

where $\overline{v}$ is the normalized velocity; $v_\alpha$ and $v_{100}$ are the actuator moving velocity under the duty cycle of $\alpha$ and that under the duty cycle of 100%, separately.

To compare the actuator moving velocity of one engaging solenoid valve under a certain duty cycle with that of all engaging solenoid valves under 100% duty cycle, an engaging velocity modulation percentage named $\beta$ and a disengaging velocity modulation percentage named $\gamma$ are given in Equations (12) and (13), separately.

$$\beta = \frac{v_\alpha}{v_1 + v_2} \times 100\% \tag{12}$$

where $v_1$ and $v_2$ are the actuator moving velocity using the QESV under 100% duty cycle and that using the SESV under 100% duty cycle, respectively.

$$\gamma = \frac{v_\alpha}{v_3 + v_4} \times 100\% \tag{13}$$

where $v_3$ and $v_4$ are the actuator moving velocity using the QDSV under 100% duty cycle and that using the SDSV under 100% duty cycle, respectively.

## 4. Simulation Analysis

*4.1. Numerical Simulatin of the Diaphragm Force and the Release Bearing Force*

The transmitted torque of a 430-type clutch about the diaphragm large end displacement for heavy-duty vehicles provided by China Sinotruck corporation is expressed in Equation (14). The units of $T_c$ and $x_l$ in Equation (14) are N·m and mm, respectively.

$$T_c = \mu' \left( 3464.483459 x_l - 336.521878 x_l^2 + 7.775271 x_l^3 \right) \tag{14}$$

The range of the diaphragm large end displacement is 10 mm for transmitting torque in this polynomial. The value of $\mu'$ is regarded as 0.25. The values of $D$ and $d$ are 430 mm and 240 mm. The number of $Z$ (the number of the friction faces) is 2. The calculation result of $R_e$ (the efficient radius of the friction force) is 171.99 mm according to its expression ($R_e = \left( D^3 - d^3 \right) / 3 \left( D^2 - d^2 \right)$). Then, $F_z$ (the force of the diaphragm acting on the pressure plate) can be expressed, calculated from Equation (15) according to Equations (8) and (14). The unit of $F_z$ in Equation (15) is N.

$$F_z = \frac{T_c}{\mu' Z R_e} = 10{,}071.75841328 x_l - 978.318152 x_l^2 + 22.603846 x_l^3 \tag{15}$$

The displacement of the pressure plate within range of disengaging (pressure plate lift displacement) is 2 mm provided by China Sinotruck corporation. Therefore, the displacement of engaging for transmitting torque and the displacement of disengaging for power interruption are 10 mm and 2 mm, respectively. To sum up, the travel of the diaphragm large end displacement is 12 mm during the clutch engaging and disengaging processes.

Simulation parameters about the diaphragm force are described below. The diaphragm large end displacement varies from zero to 12 mm. The range of the diaphragm large end displacement for the clutch engagement region is from zero to 10 mm, and that for the clutch disengagement region is from 10 mm to 12 mm. The displacement interval is 0.01 mm. Writing a MATLAB script program according to Equation (15), based on every given diaphragm large end displacement according to the displacement interval, the diaphragm force is calculated. In total, 1000 sets of data within engaging range and

200 sets of data within disengaging range are provided to plot the simulation curve of the diaphragm force in MATLAB.

The simulation curve of the diaphragm force acting on the pressure plate based on MATLAB is shown in Figure 6. Figure 6a shows the variation of the diaphragm force within the engaging range, such that the maximum value of the diaphragm force is 30,362 N when the diaphragm large end displacement is 6.67 mm. Figure 6b shows the variation of the diaphragm force within the disengaging range such that the minimum value of the diaphragm force is 19,043 N when the diaphragm large end displacement is 12 mm.

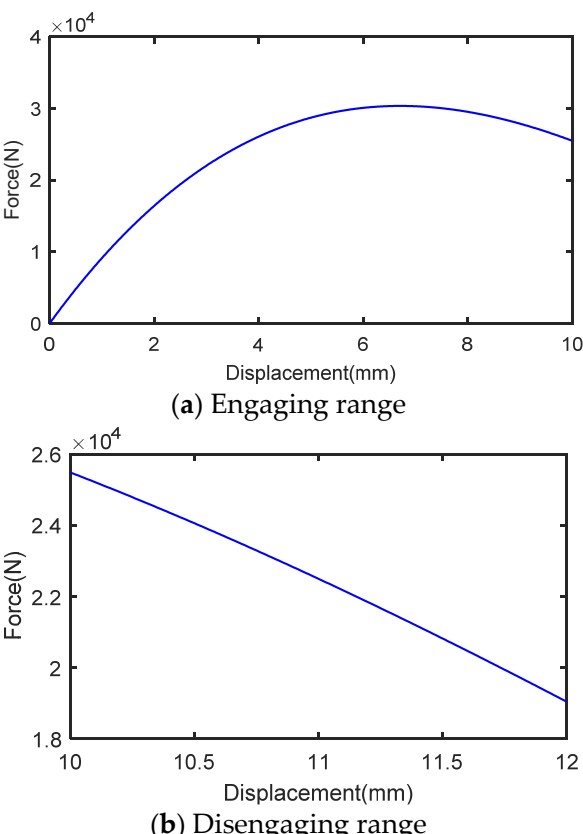

(**a**) Engaging range

(**b**) Disengaging range

**Figure 6.** Simulation curve of the diaphragm force.

The lever ratios and the release bearing travel displacement provided by the China Sinotruck corporation are described as follows. The lever ratio of the diaphragm from the small end to the diaphragm large end is 1.35 ($k_d$ = 1.35). The lever ratio of the release fork from the piston rod to the release bearing is 1.71 ($k_r$ = 1.71). The release bearing empty stroke displacement is designed to eliminate the gap between the release bearing and the diaphragm small end. Its empty stroke displacement is 6.90 mm. To describe the release bearing force limited by the diaphragm large end, the release bearing displacement is divided into two parts: the force loading region is 16.20 mm, corresponding to the clutch disengaging travel displacement (12 mm), and the no force loading region is 6.90 mm, corresponding to empty stroke displacement. The release bearing force about its displacement from the beginning of the clutch disengaging process is calculated from Equation (16) according to Equation (15). The units of $F_b$ and $x_b$ are N and mm, respectively.

$$\begin{cases} F_b = 5526.342066x_b - 397.629692x_b^2 + 6.805301x_b^3 & (0 \le x_b \le 16.20) \\ F_b = 0 & (16.20 < x_b \le 23.10) \end{cases} \quad (16)$$

Simulation parameters are described below. The release bearing displacement varies from zero to 23.10 mm. The displacement interval is 0.01 mm. Writing the MATLAB script program according to Equation (16), the release bearing force can be calculated. The

simulation curve of the release bearing force about the release bearing displacement based on MATLAB during the disengaging process is shown in Figure 7. The maximum value of the release bearing force is 22,491 N when the release bearing displacement is 9.03 mm.

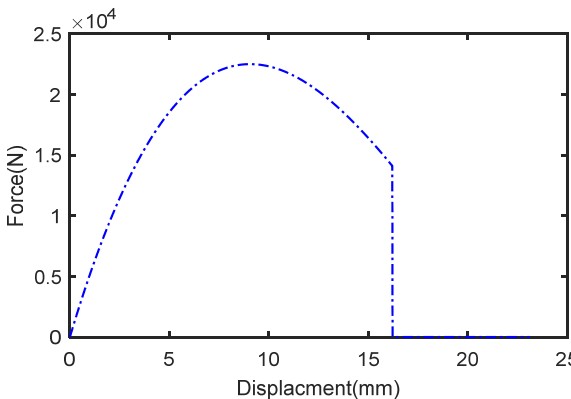

**Figure 7.** Simulation curve of the release bearing force during the disengaging process.

### 4.2. Simulation Analysis of the Clutch Actuator

A simulation model of PCA was built based on AMESim simulation platform and shown in Figure 8. To prove the validity of the motion analyzing in Section 2 above, the simulation results of engaging process using the QESV and disengaging process using the QDSV are analyzed. The simulation results of engaging process using SESV and disengaging process using SDSV are not shown in this section because the curve trends are similar to those using QESV and QESV.

Based on Equations (7), (10) and (16), some parameters are given. The orifice areas of the QDSV and the QESV are 30 mm². The piston and rod diameters are 120 mm and 12 mm. The damping coefficient of the piston moving is 5 N/(m/s). The elastic coefficient of the cylinder spring is 10,000 N/m. The total mass being moved is 50 kg. The air source is realistic dry air provided by AMESim system. Corresponding to the release bearing displacement, the empty stroke displacement of the piston is from 27.70 mm to 39.50 mm. The disengaging range of the piston displacement is from 0 to 27.70 mm, which is related to the clutch disengaging travel displacement (12 mm). Therefore, the release bearing force can be calculated from Equation (16).

The simulation curve of the actuator displacement using the QESV under different duty cycles is shown in Figure 9. It shows that the time of the engaging process increases with the decreasing of the duty cycle. The times of the engaging process are 6.55 s, 4.20 s, 2.80 s, 2.05 s, 1.67 s, 1.47 s, and 1.26 s under the duty cycles of 30%, 40%, 50%, 60%, 70%, 80%, and 100% respectively. The average velocity under a 30% duty cycle is about 19.24% of that under a 100% duty cycle. At the beginning of the simulation, the piston moves fast owing to the clutch empty stroke displacement of 11.80 mm (39.50 − 27.70 = 11.80 mm). With the releasing of the pressure air in the piston chamber, the piston moves more slowly. The reasons mainly include two aspects, one is the larger release bearing force from the diaphragm force, the other one is the lower pressure of the piston chamer.

Similarly, the simulation curve of actuator displacement using the QDSV under different duty cycles is shown in Figure 10. It is shown that the time of the disengaging process declines with the increasing of the duty cycle. The times of the disengaging process are 0.55 s, 0.57 s, 0.63 s, 0.75 s, 0.81 s, 0.91 s, and 1.18 s under duty cycles of 30%, 40%, 50%, 60%, 70%, 80%, and 100%, respectively. The average velocity under a 30% duty cycle is about 46.61% of that under a 100% duty cycle. With declines in the duty cycle, distortion phenomena are shown because the clutch disengaging process needs larger force to overcome the diaphragm force to separate the clutch disk and the pressure plate.

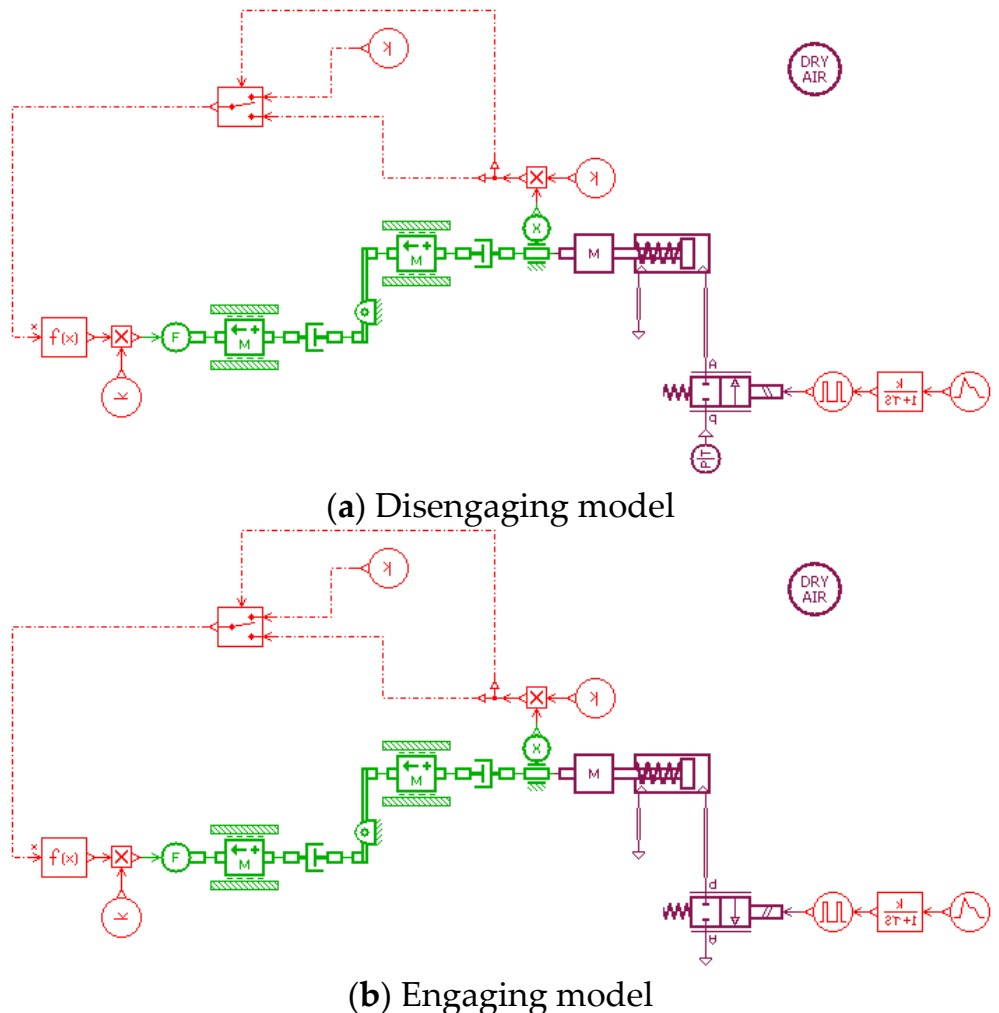

(**a**) Disengaging model

(**b**) Engaging model

**Figure 8.** Simulation model of a clutch actuator based on AMESim.

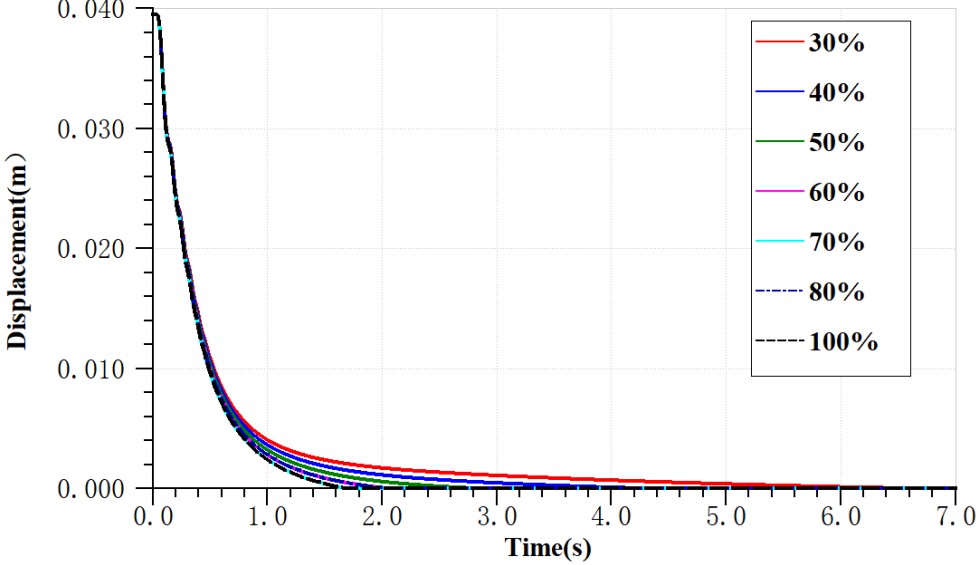

**Figure 9.** Simulation curve of actuator displacement using the QESV.

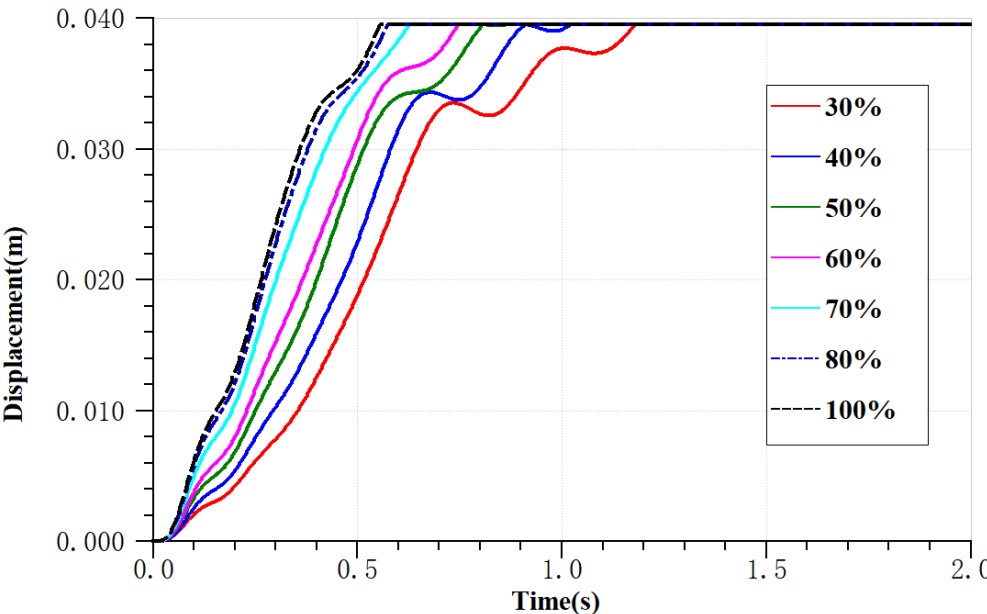

**Figure 10.** Simulation curve of the actuator displacement using the QDSV.

## 5. Test

The AMT test bench in the paper is shown in Figure 11. The direction of the displacement is determined such that zero is defined as the full engaged position, and the position at 39.50 mm is defined as the full disengaged position. The motion characteristics of the clutch actuator can be analyzed by controlling the solenoid valves. The frequencies of the solenoid valves are 100 HZ. The rated voltage of every solenoid valve is DC 24 V. The pressure of the pneumatic system is 0.8 MPa. The AMT is made by China Sinotruck corporation. The solenoid valve is almost closed because of small driving power if the duty cycle is less than 30%. Thus, the actual range of the duty cycle changes from 30% to 100%.

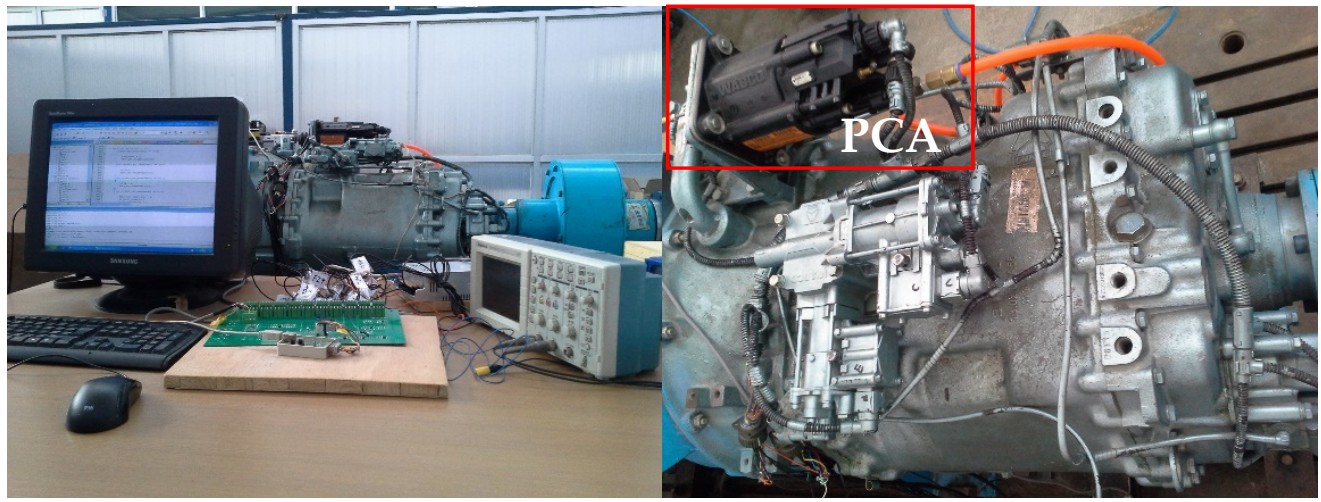

**Figure 11.** AMT test bench.

### 5.1. Motion Characteristics during the Engaging Process

5.1.1. Motion Characteristics Using the QESV

The test data are achieved using the QESV alone, while the other solenoid valves of PCA are closed. The actuator displacement-time curve and the actuator velocity-displacement curve are obtained using the QESV under different duty cycles in Figures 12 and 13, respectively.

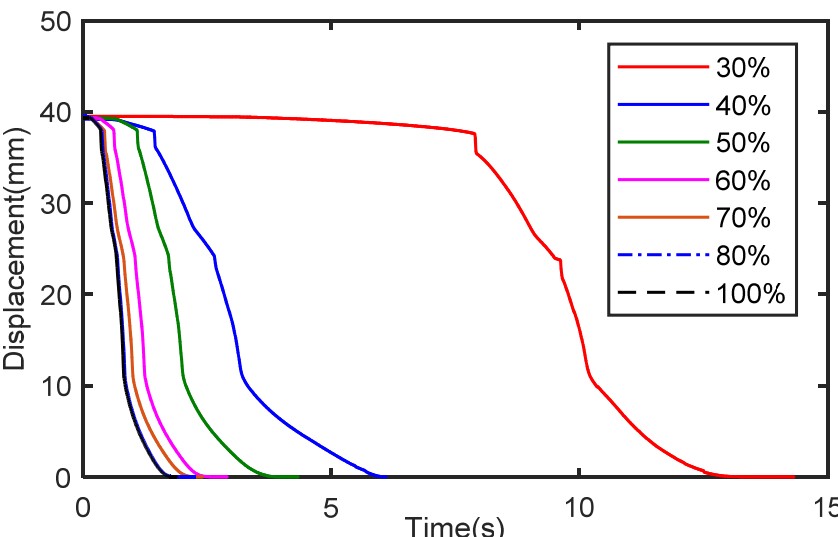

**Figure 12.** Actuator displacement-time curve using the QESV under different duty cycles.

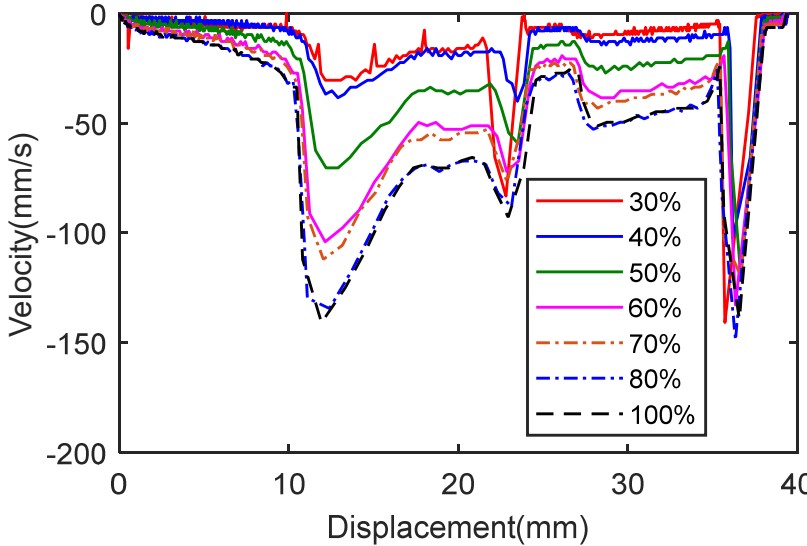

**Figure 13.** Actuator velocity-displacement curve using the QESV under different duty cycles.

As is shown in Figure 12, the lag times of the cylinder operation using the QESV under duty cycles of 30%, 40%, 50%, 60%, 70%, 80%, and 100% are 2500 ms, 250 ms, 120 ms, 100 ms, 70 ms, 60 ms, and 60 ms. Obviously, the lag time decreases with the increasing of the duty cycle. The displacement-time curves under 80% and 100% duty cycles are almost coincident, which indicates that the QESV under an 80% duty cycle is nearly in a condition of full opening.

Figure 13 shows the actuator velocity is relatively small at the beginning because of the lag times of QESV and the clutch cylinder. The actuator average engaging velocity increases with the increasing of the duty cycle of the QESV. It is shown that the velocity-displacement curves have the same tendency that increases first and then decreases during the motion from the position of 20 mm to the position of 5 mm. The peak of the actuator velocity is 140.80 mm per second under 100% duty cycle at the position of 11.94 mm, whereas it is 30.40 mm per second under 30% duty cycle at the position of 13.96 mm.

The actuator normalized velocity using the QESV under different duty cycles is calculated in Table 1. Table 1 shows that the actuator average engaging velocity slows down when the duty cycle of the QESV is less than 60%. The range of the actuator normalized velocity varies from 13.28% to 100%.

**Table 1.** Normalized velocities using the QESV.

| Duty (%) | Displacement (mm) | Time (s) | Average Velocity (mm/s) | Normalized Velocity (%) |
|---|---|---|---|---|
| 30 | 39.50 | 14.33 | 2.76 | 13.28 |
| 40 | 39.50 | 6.11 | 6.46 | 31.07 |
| 50 | 39.50 | 4.34 | 9.10 | 43.77 |
| 60 | 39.50 | 2.91 | 13.57 | 65.27 |
| 70 | 39.50 | 2.41 | 16.39 | 78.84 |
| 80 | 39.50 | 2.27 | 17.40 | 83.69 |
| 100 | 39.50 | 1.90 | 20.79 | 100 |

## 5.1.2. Motion Characteristics Using the SESV

The test data are gained using the SESV alone while the other solenoid valves of PCA are closed. The actuator displacement-time curve and the actuator velocity-displacement curve are received using the SESV under different duty cycles in Figures 14 and 15 separately.

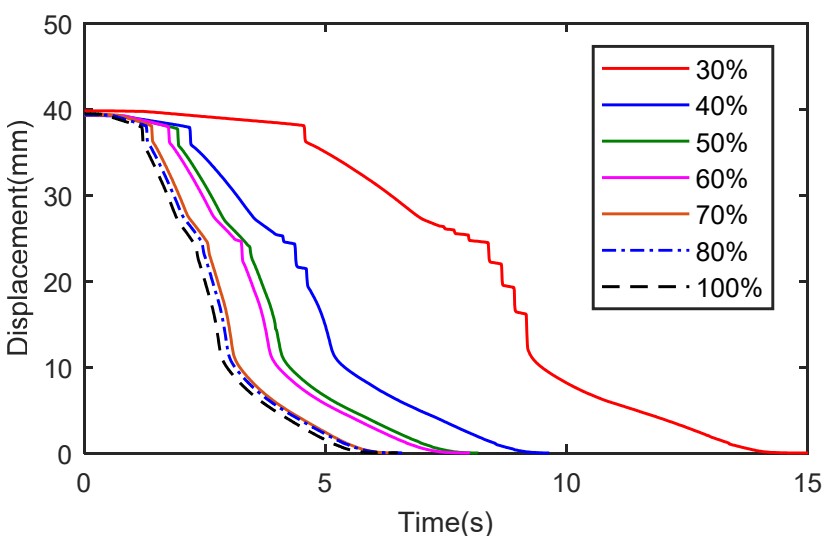

**Figure 14.** Actuator displacement-time curve using the SESV under different duty cycles.

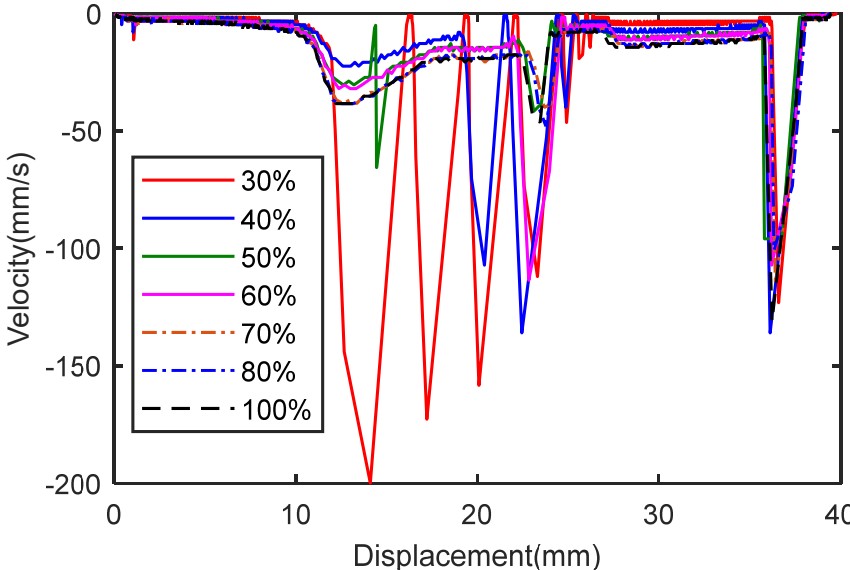

**Figure 15.** Actuator velocity-displacement curve using the SESV under different duty cycles.

Figure 14 shows that the lag times of the cylinder operation using the SESV under the duty cycles of 30%, 40%, 50%, 60%, 70%, 80%, and 100% are 610 ms, 220 ms, 180 ms, 150 ms, 130 ms, 110 ms, and 110 ms. The lag time using SESV decreases with the increasing of the duty cycle, which is similar to that using QESV. The displacement-time curves under the duty cycles of 70%, 80%, and 100% are almost identical, which indicates that the SESV under 70% duty cycle is nearly in a full opening condition. The displacement-time curves under the duty cycles of 30% and 40% have some distortion phenomena from the position of 27 mm to the position of 12 mm, which will result in the displacement distortion of the diaphragm large end. According to Equations (6) and (8), the force of the pressure plate acting on the clutch disk and the clutch-transmitted torque will have sudden changes, which will result in transmission shocks. Considering the purpose of reducing shocks, a slower engaging velocity is needed from the position of 20 mm to 5 mm for the half-engagement point. Thus, if the SESV is used alone, the duty cycle of less than 50% should be avoided from the position of 20 mm to the position of 12 mm.

Figure 15 shows that the actuator velocity is relatively small at the beginning because of the lag times of SESV and the clutch cylinder. The actuator moving velocity changes slowly when the duty cycle is less than 50%. The actuator engaging velocity under 30% duty cycle shows drastic shocks from the position of 27 mm to the position of 12 mm, and the maximum velocity reaches 200 mm per second, whereas the maximum of the actuator engaging velocity under 100% duty cycle is only 38.40 mm per second within the same displacement range. This finding means the actuator engaging velocity under a 30% duty cycle changes dramatically and should be paid special attention during the clutch engaging process.

The actuator normalized velocities using the SESV under different duty cycles is calculated in Table 2. The normalized velocity varies from 42.86% to 100% using the SESV by PWM signals. The average engaging velocity varies from 2.61 mm per second to 6.09 mm per second. Considering the precise control for the half-engagement point, the range of the engaging velocity using the SESV should be mainly considered during the starting process.

**Table 2.** Normalized velocities using the SESV.

| Duty (%) | Displacement (mm) | Time (s) | Average Velocity (mm/s) | Normalized Velocity (%) |
|---|---|---|---|---|
| 30 | 39.50 | 15.11 | 2.61 | 42.86 |
| 40 | 39.50 | 9.63 | 4.10 | 72.41 |
| 50 | 39.50 | 8.17 | 4.83 | 79.31 |
| 60 | 39.50 | 7.99 | 4.94 | 81.12 |
| 70 | 39.50 | 6.58 | 6.00 | 98.52 |
| 80 | 39.50 | 6.50 | 6.08 | 99.84 |
| 100 | 39.50 | 6.49 | 6.09 | 100 |

### 5.1.3. Motion Characteristics Using a Combination of the QESV and the SESV

Figures 12 and 14 shows a nonlinear relationship between the actuator displacement and the time, which is related to several nonlinear factors such as the performances of the diaphragm, the structure of the solenoid valve, the structure of the cylinder, and the pressure of compressed air. Tables 1 and 2 show that the actuator average engaging velocity using the SESV alone is 6.09 mm per second, that using the QESV alone is 20.79 mm per second, and the former accounts for 29.29% of the latter. The range of the normalized velocity using the QESE alone is clearly larger than that using the SESV alone.

The actuator displacement-time curve and the actuator velocity-displacement curve using the QESV and the SESV under a duty cycle of 100% are shown in Figures 16 and 17, respectively. The engaging motion time is 1.47 s. The average engaging velocity is 26.87 mm per second that is nearly equal to the sum of the average engaging velocities using every engaging solenoid valve with 100% duty cycle, which means controlling the duty cycle is to control the valve flow. The maximum engaging velocity is 180.80 mm per second at

the position of 11.57 mm in Figure 17, and it is less than the maximum velocity of 200 mm per seconds using the SESV in Figure 15. Therefore, using the SESV with a lower duty cycle is a difficult problem for reducing the shocks that require mastering the actuator motion characteristics.

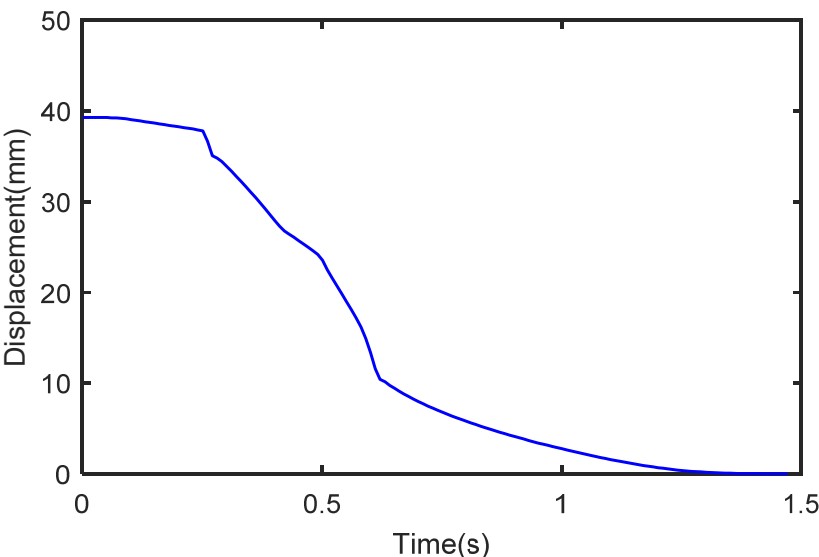

**Figure 16.** Actuator displacement-time curve using the QESV and the SESV under a 100% duty cycle.

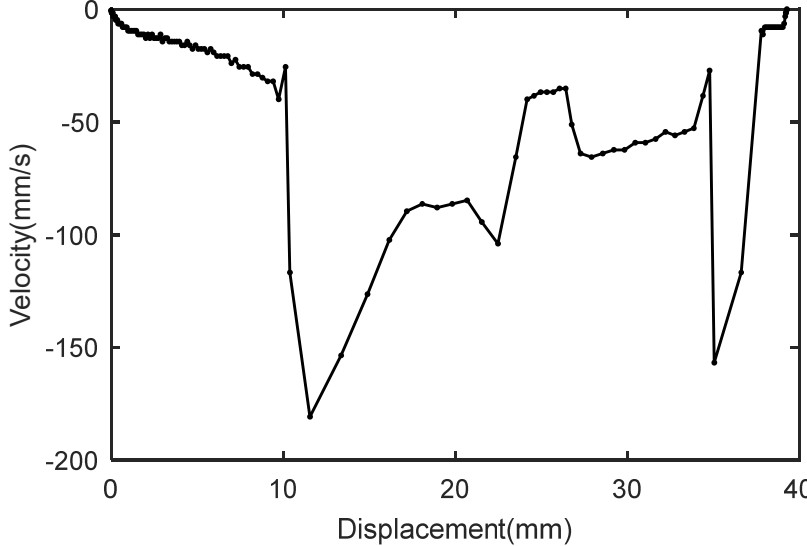

**Figure 17.** Actuator velocity-displacement curve using the QESV and the SESV under a 100% duty cycle.

The lag time of the actuator is approximately 60 ms using the QESV under a 100% duty cycle in Figure 12 and 110 ms using the SESV under a 100% duty cycle in Figure 14, whereas it is only 30 ms using the two engaging solenoid valves under 100% duty cycles shown in Figure 16. Obviously, the lag time decreases because of large compressed air flow. Opening the valve needs a longer time if the valve flow becomes smaller, which can be supported from Equations (3) and (10).

Based on Equation (12), the engaging velocity modulation percentages are shown in Table 3. The engaging velocity modulation percentage for the QESV varies from 10.27% to 77.37% and that for the SESV varies from 9.71% to 22.66%.

**Table 3.** Engaging modulation velocity percentage.

| Valve | Duty (%) | Average Velocity (mm/s) | Velocity Modulation Percentage (%) |
|---|---|---|---|
| QESV | 30 | 2.76 | 10.27 |
| | 40 | 6.46 | 24.04 |
| | 50 | 9.10 | 33.87 |
| | 60 | 13.57 | 50.50 |
| | 70 | 16.39 | 61.00 |
| | 80 | 17.40 | 64.76 |
| | 100 | 20.79 | 77.37 |
| SESV | 30 | 2.61 | 9.71 |
| | 40 | 4.10 | 15.26 |
| | 50 | 4.83 | 17.98 |
| | 60 | 4.94 | 18.38 |
| | 70 | 6.00 | 22.33 |
| | 80 | 6.08 | 22.63 |
| | 100 | 6.09 | 22.66 |
| QESV and SESV | 100 | 26.87 | 100 |

The relationship between the engaging velocity and the duty cycle can be approximately calculated in Equation (17).

$$v_c = v_{\alpha 1} + v_{\alpha 2} = v_1 \beta_1 + v_2 \beta_2 \tag{17}$$

where $\alpha 1$ is the duty cycle of the QESV; $\alpha 2$ is the duty cycle of the SESV; $\beta_1$ is the engaging velocity modulation percentage under the duty of $\alpha 1$; $\beta_2$ is the engaging velocity modulation percentage under the duty of $\alpha 2$; $v_{\alpha 1}$ is the actuator engaging velocity the duty of $\alpha 1$, and $v_{\alpha 2}$ is the actuator engaging velocity, the duty of $\alpha 2$.

*5.2. Motion Characteristics during the Disengaging Process*

5.2.1. Motion Characteristics Using the QDSV

The test data are obtained only using the QDSV, while the other solenoid valves of PCA are closed. The actuator displacement-time curve and the velocity-displacement curve using the QDSV under different duty cycles are shown in Figures 18 and 19 separately.

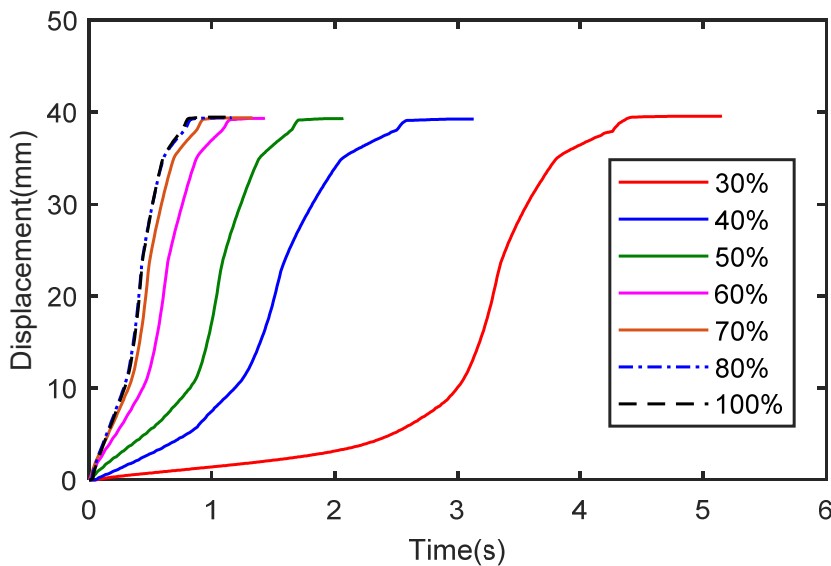

**Figure 18.** Actuator displacement-time curve using the QDSV under different duty cycles.

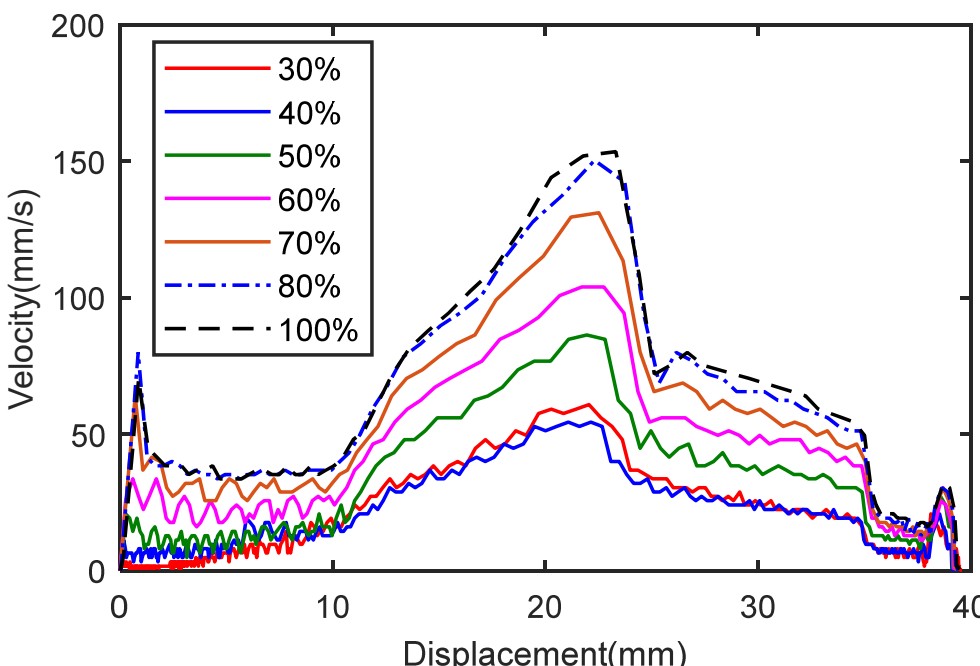

**Figure 19.** Actuator velocity-displacement curve using the QDSV under different duty cycles.

It is shown in Figure 18 that the lag times of the cylinder operation using the QDSV under the duty cycles of 30%, 40%, 50%, 60%, 70%, 80%, and 100% are 60 ms, 50 ms, 30 ms, 20 ms, 10 ms, 10 ms, and 10 ms respectively. The lag time using QDSV decreases with the increasing of the duty cycle, which is similar to that using QESV or SESV. To sum up, the lag time of QDSV in Figure 18 is obviously less than that of QESV in Figure 12. It is designed for the purpose of quick disengaging for the clutch. The actuator displacement-time curves under the duty cycles of 80% and 100% are almost identical, which shows that the QDSV under an 80% duty cycle is nearly in a condition of full opening. Figure 19 shows that the maximum disengaging velocity is 153.60 mm per second using the QDSV under a 100% duty cycle. Table 4 shows that the actuator normalized velocity using the QDSV under different duty cycles varies from 22.53% to 100%.

**Table 4.** Normalized velocities using the QDSV.

| Duty (%) | Displacement (mm) | Time (s) | Average Velocity (mm/s) | Normalized Velocity (%) |
|---|---|---|---|---|
| 30 | 39.50 | 5.15 | 7.67 | 22.53 |
| 40 | 39.50 | 3.13 | 12.62 | 37.06 |
| 50 | 39.50 | 2.07 | 19.08 | 56.04 |
| 60 | 39.50 | 1.43 | 27.62 | 81.12 |
| 70 | 39.50 | 1.33 | 29.70 | 87.22 |
| 80 | 39.50 | 1.18 | 33.47 | 98.30 |
| 100 | 39.50 | 1.16 | 34.05 | 100 |

The actuator normalized velocity using the QDSV under different duty cycles is listed in Table 4. The actuator normalized velocity varies from 22.53% to 100%.

### 5.2.2. Motion Characteristics Using the SDSV

The test data are obtained using the SDSV alone, while the other solenoid valves of PCA are closed. The actuator displacement-time curve and the velocity-displacement curve using the SDSV under different duty cycles are shown in Figures 20 and 21 separately.

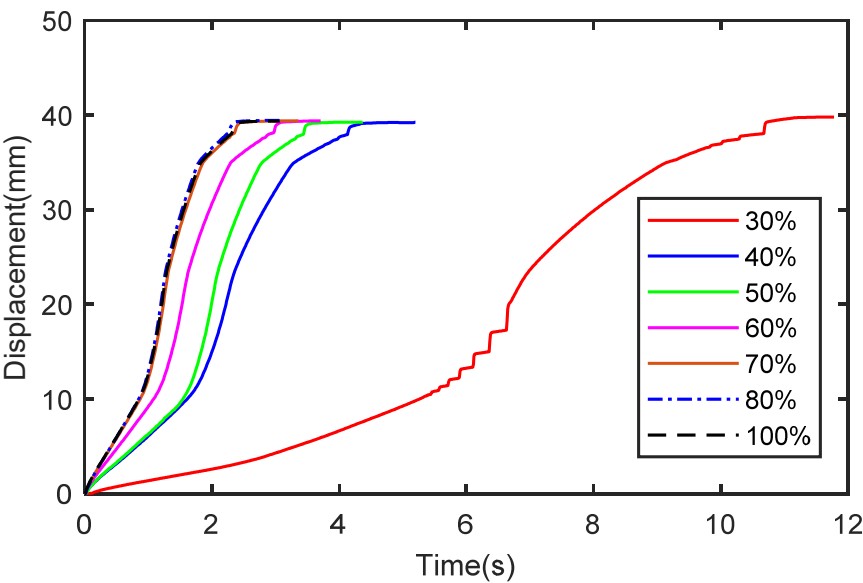

**Figure 20.** Actuator displacement-time curve using the SDSV under different duty cycles.

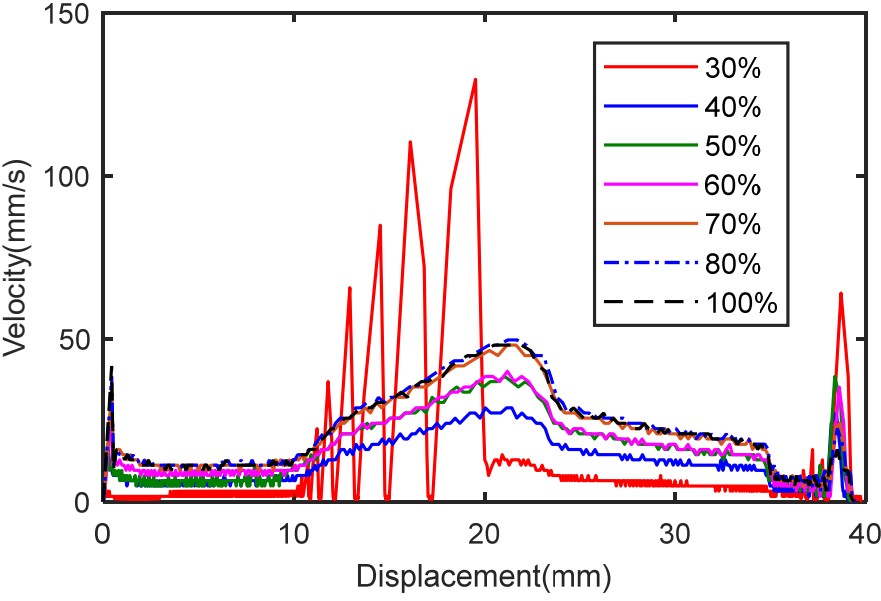

**Figure 21.** Actuator velocity-displacement curve using the SDSV under different duty cycles.

It is shown in Figure 20 that the lag times of the cylinder operation using the SDSV under the duty cycles of 30%, 40%, 50%, 60%, 70%, 80%, and 100% are 90 ms, 30 ms, 30 ms, 20 ms, 20 ms, 20 ms, and 20 ms, respectively. The lag time using the SDSV decreases with the increasing of the duty cycle, which is similar to that using the QDSV. The actuator displacement-time curves under the duty cycles of 70%, 80%, and 100% are nearly identical, which indicates that the SDSV under a 70% duty cycle is almost in a condition of full opening. The displacement-time curve under a 30% duty cycle has some distortion phenomena from the position of 10 mm to the position of 20 mm, which will result in the displacement distortion of the diaphragm large end and transmission shock. Thus, the duty cycle of the SDSV is not advised to be used less than 40% from the position of 10 mm to the position of 20 mm.

Figure 21 shows the relationship between the velocity and the displacement under different duty cycles of the SDSV. The actuator disengaging velocity under 30% duty cycle shows drastic shocks from the position of 10 mm to the position of 20 mm in Figure 16,

and the maximum velocity reaches 129.60 mm per second, whereas the maximum of the actuator disengaging velocity under a 100% duty cycle is only 44.80 mm per second within the same displacement range. The distortion is caused by the force of the diaphragm spring, and the force increases with the increasing of the displacement within this range, which can be deduced in Equation (6). The pressure and the compressed air flow cannot provide nearly enough force acting on the piston to pull the release bearing easily, and the pressure plate cannot move away from the clutch disk easily, which can be explained from Figure 6. Therefore, the duty cycle should be selected to be larger so as to overcome the force of the diaphragm spring easily. It shows that the maximum of the actuator disengaging velocity is 49.60 mm per second using the SDSV under a 100% duty cycle, whereas it is 129.60 mm per second using the SDSV under a 30% duty cycle. Thus, the lower duty cycle of the SDSV is not advisable for use within this displacement range (from the position of 10 mm to the position of 20 mm).

The actuator normalized velocity using the SDSV under different duty cycles in Table 5 varies from 25.95% to 100%.

**Table 5.** Actuator normalized velocities using the SDSV.

| Duty (%) | Displacement (mm) | Time (s) | Average Velocity (mm/s) | Normalized Velocity (%) |
|---|---|---|---|---|
| 30 | 39.50 | 11.78 | 3.35 | 25.95 |
| 40 | 39.50 | 5.20 | 7.60 | 58.87 |
| 50 | 39.50 | 4.37 | 9.04 | 70.02 |
| 60 | 39.50 | 3.71 | 10.65 | 82.49 |
| 70 | 39.50 | 3.36 | 11.76 | 91.09 |
| 80 | 39.50 | 3.10 | 12.74 | 98.68 |
| 100 | 39.50 | 3.06 | 12.91 | 100 |

### 5.2.3. Motion Characteristics Using a Combination of the QDSV and the SDSV

To analyze the motion characteristics for using a combination of the QDSV and the SDSV further, the actuator displacement-time curve and the actuator velocity-displacement curve using the QDSV and the SDSV under a duty cycle of 100% are shown in Figures 22 and 23 separately.

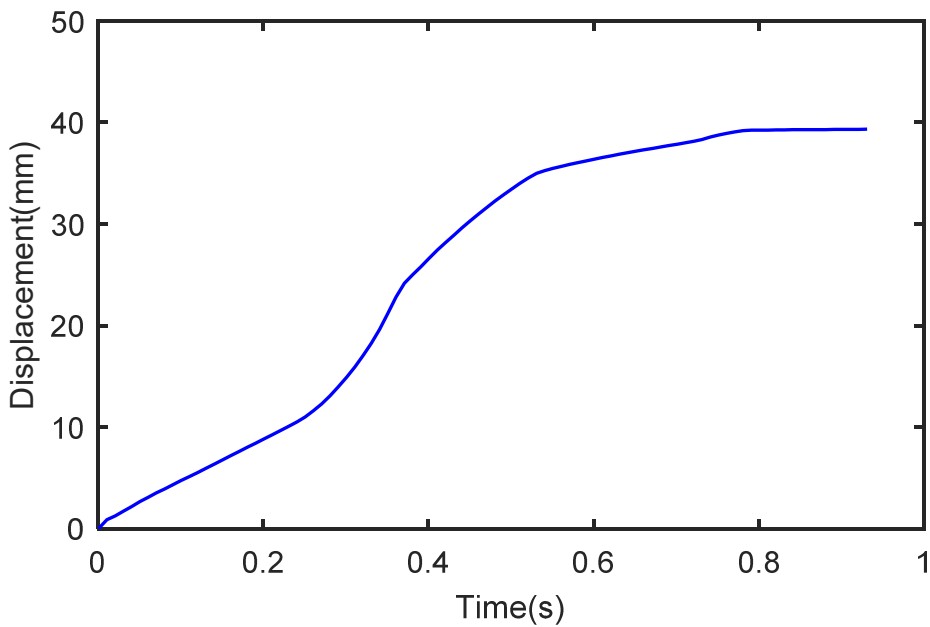

**Figure 22.** Actuator displacement-time curve using the QDSV and the SDSV under a 100% duty cycle.

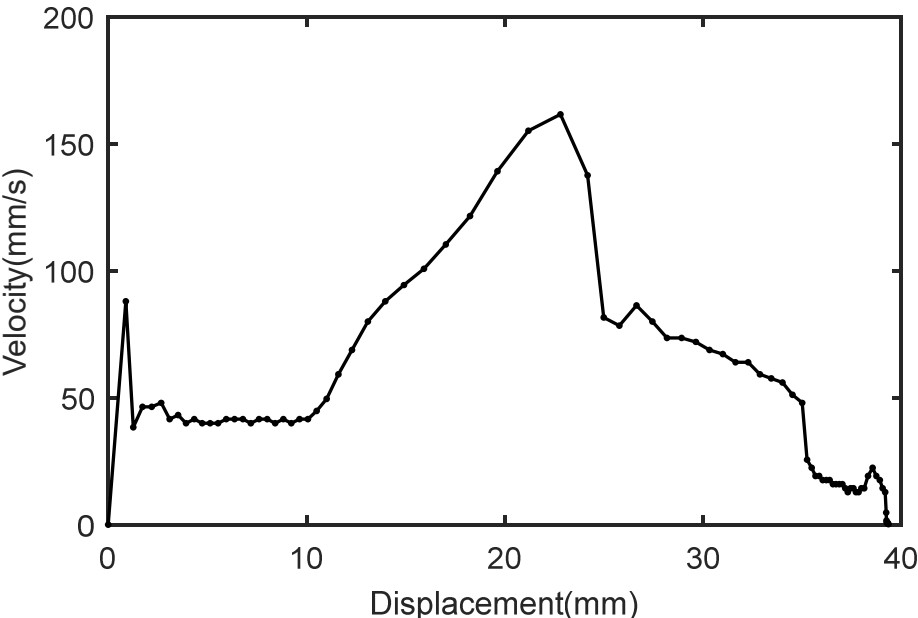

**Figure 23.** Actuator velocity-displacement curve using the QDSV and the SDSV under a 100% duty cycle.

The lag time is approximately 60 ms using the QDSV under a100% duty cycle as shown in Figure 18 and 90 ms using the SDSV under 100% duty cycle as shown in Figure 19, whereas it is only 10 ms using the two disengaging solenoid valves under 100% duty cycles in Figure 22. Clearly, increasing the valve flow can reduce the startup time of the actuator. Figure 23 shows that the maximum of the actuator disengaging velocity is 161.60 mm per second at the position of 22.80 mm.

Based on Equation (13), the disengaging velocity modulation percentages are shown in Table 6. The average disengaging velocity using the two solenoid valves under a 100% duty cycle is 42.47 mm per second, and it is nearly equal to the sum of the actuator velocity using the QDSV alone under a 100% duty cycle and that using the SDSV alone under a 100% duty cycle.

**Table 6.** Disengaging velocity modulation percentage.

| Valve | Duty (%) | Average Velocity (mm/s) | Velocity Modulation Percentage (%) |
|---|---|---|---|
| QDSV | 30 | 7.67 | 18.06 |
| | 40 | 12.62 | 29.71 |
| | 50 | 19.08 | 44.93 |
| | 60 | 27.62 | 65.03 |
| | 70 | 29.48 | 69.41 |
| | 80 | 33.47 | 78.81 |
| | 100 | 34.05 | 80.17 |
| SDSV | 30 | 3.35 | 7.89 |
| | 40 | 7.60 | 17.90 |
| | 50 | 9.04 | 21.29 |
| | 60 | 10.65 | 25.08 |
| | 70 | 11.76 | 27.69 |
| | 80 | 12.74 | 30.00 |
| | 100 | 12.91 | 30.40 |
| QDSV and SDSV | 100 | 42.47 | 100 |

Table 6 shows that the disengaging velocity percentage using the QDSV alone by PWM signals varies from 18.06% to 80.17%, and that using the SDSV alone by PWM signals

varies from 7.89% to 30.40%. The maximum disengaging velocity percentage using the SDSV accounts for 37.92% of that using the QDSV.

The relationship between the actuator disengaging velocity and the duty cycle can be approximated in Equation (18).

$$v_c = v_{\alpha 3} + v_{\alpha 4} = v_3 \beta_3 + v_4 \beta_4 \tag{18}$$

where $\alpha 3$ is the duty cycle of the QDSV; $\alpha 4$ is the duty cycle of the SDSV; $\beta_3$ is the disengaging velocity modulation percentage under the duty of $\alpha 3$; $\beta_4$ is the disengaging velocity modulation percentage under the duty of $\alpha 4$; $v_{\alpha 3}$ is the actuator disengaging velocity under the duty of $\alpha 3$, and $v_{\alpha 4}$ is the actuator disengaging velocity under the duty of $\alpha 4$.

## 6. Brief Descriptions of Actuator Motion Characteristics

Some motion characteristics of the clutch actuator can be achieved by simulation analysis and test results. Simulation analysis proves the basic validity of clutch motion analysis (Section 2). Simulation and test curves have same trends about the engaging processes in Figures 9 and 12. The same trends about the disengaging processes in Figures 10 and 18 are also similar. The deficiencies of the simulation cannot reveal the actual operation conditions in terms of pressure impact, gas leak, manufacture tolerance, and mechanical wear. The actions of the actuator change with the variations in the duty cycle. The less the duty cycle is, the slower the piston moves. The lag actions of the actuator in tests are slower than those in simulation. Essentially, the simulation results provide the main principle of dynamics. Consequently, the test results provide the actual clutch motion characteristics for the purpose of better control.

The clutch actuator engaging characteristics are briefly described below. The lag time of the actuator operation decreases with the increasing of the duty cycle of the QESV or the SESV. The actuator average engaging velocity increases with the increasing of the duty cycle of the QESV or the SESV. The QESV under a duty of 80% and the SESV under a duty of 70% are almost considered in a condition of full opening position. As for the QESV, the actuator normalized velocity varies from 13.28% to 100%, and the actuator velocity modulation percentage varies from 10.27% to 77.37%. As for the SESV, the actuator normalized velocity varies from 42.86% to 100%, and the actuator velocity modulation percentage varies from 9.71% to 22.66%. In combination with the two engaging solenoid valves, the actuator velocity modulation percentage varies from 9.71 to 100% and can meet the requirements of various engaging velocities for heavy-duty vehicles to start up and shift smoothly. Particularly, the actuator engaging velocity can be estimated as the sum of the actuator engaging velocity using the QESV alone and that using the SESV alone. If an engaging solenoid valve is used under a certain duty cycle, the actuator engaging velocity is equal to the product of the actuator engaging velocity under a 100% duty cycle and this duty cycle. The phenomenon of the displacement distortion will lead to transmission shocks and vibrations for heavy-duty vehicles if the SESV is only used under a duty cycle of less than 50%. So, the duty cycle of the SESV should not be less than 50% during the clutch engaging process from the position of 20 mm to the position of 5 mm.

Moreover, the clutch actuator disengaging characteristics can be briefly summarized. The lag time of the actuator operation decreases with the increasing of the duty cycle of the QDSV or the SDSV. The actuator average disengaging velocity increases with the increasing of the duty cycle. The QDSV under a duty of 80% and the SDSV under a duty of 70% are considered in a condition of full opening position. As regards to the QDSV, the range of the actuator normalized velocity changes from 22.53% to 100%, and the range of the actuator velocity modulation percentage changes from 18.06% to 80.17%. As regards to the SDSV, the range of the actuator normalized velocity varies from 25.95% to 100%, and the range of the actuator velocity modulation percentage varies from 7.89% to 100%. Combining the two disengaging solenoid valves, the actuator velocity modulation percentage varies from 7.89% to 100%, which can meet the requirements of various disengaging velocities within this scope. Particularly, the actuator velocity can be estimated as the sum of the

actuator disengaging velocity using the QDSV alone and that using the SDSV alone. The actuator disengaging velocity using a disengaging solenoid valve under a certain duty cycle equals the product of the disengaging velocity under a 100% duty cycle and this duty cycle. Distortions of the displacement-time curve and the velocity-displacement curve are shown from the position of 10 mm to the position of 20 mm when the SDSV is used alone under the duty of less than 40%. Therefore, the duty cycle of the SDSV should not be less than 40% when it is used alone during the clutch disengaging process from the position of 10 mm to the position of 20 mm.

## 7. Conclusions

Based on the clutch modeling and analyzing, simulation analysis about the clutch actuator system is studied to verify that the clutch modeling is effective and reasonable. In a test study, normalized velocity and velocity modulation percentage are proposed as the main evaluation parameters if the clutch actuator is controlled by PWM signals. Based on the experiments, the motion characteristics of the clutch actuator are obtained. For every solenoid valve, the lag time of the actuator operation decreases with the increasing of the duty cycle, and the actuator average velocity increases with the increasing of the duty cycle. By analyzing the displacement-time curves and the velocity-displacement curves, the range of the actuator normalized velocities and the range of the actuator velocity modulation percentages are obtained for the clutch-engaging process and the clutch-disengaging process. By comparison and analysis, the estimation expressions of actuator engaging velocity and actuator disengaging velocity are put forward when the solenoid valves are used in combination with PWM signals. For the sake of reducing transmission shocks and improving smoothness for heavy-duty vehicles, the duty cycles of the SESV and the SDSV are provided. The experimental data provide some necessary evidence for the better control of AMT in heavy-duty vehicles.

**Author Contributions:** Conceptualization, Y.L. and Z.W.; methodology, Y.L.; software, Y.L.; validation, Y.L. and Z.W.; formal analysis, Z.W.; investigation, Z.W.; resources, Y.L.; data curation, Y.L.; writing—original draft preparation, Y.L.; writing—review and editing, Y.L.; visualization, Y.L.; supervision, Y.L.; project administration, Y.L.; funding acquisition, Z.W. Both authors have read and agreed to the published version of the manuscript.

**Funding:** This work was supported by the Natural Science Foundation of Shandong Province, China (Grant No. ZR2018MEE015).

**Institutional Review Board Statement:** Not applicable.

**Informed Consent Statement:** Not applicable.

**Data Availability Statement:** Data sharing not applicable.

**Conflicts of Interest:** The authors declare no conflict of interest.

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
