# Peer review of "Motion Characteristics of a Clutch Actuator for Heavy-Duty Vehicles with Automated Mechanical Transmission"

_actuators, doi:10.3390/act10080179_

Round 1

Reviewer 1 Report

The manuscript deals with the study of a pneumatic clutch actuator for heavy-duty vehicles. A mathematical description of the actuator behaviour is proposed and an experimental analysis is carried out. Presented results are of some interest and worth to be published. However, in my opinion the contribution is not introduced adequately. In particular it is not clear what is the role of the actuator modelling, i.e., what described in Section 2.2. My concern is twofold: 1) the overall model appears not to be complete since the dependence of the intensive variable p (compressed air pressure) on external forcing variables is not indicated 2) it appears that the dynamic model of the actuator is not used: neither the comparison between experiments and model analysis, nor simulation of the model have been carried out.

Thus I am wondering which is the aim of the whole Section 2.2 in the context of the manuscript.

Below few minor comments:

  • I would suggest to add physical variables that appear in the model of Sect. 2.2 in Figure 1 (or a further figure to be added) so to clarify all equations (e.g., x_b, x_c, x_v, p might be indicated in some figure)      
  • Figs. 6 and 8 should have the same x-scale so to be compared easily
  • The bibliography analysis should be improved since it is too much biased towards references available only in Chinese language and some highly cited English papers in the field have not been referenced.   

Reviewer 2 Report

This paper studies the motion characteristics of a clutch actuator for heavy-duty vehicles with AMT, and presents the concept of normalized velocity and velocity modulation percentage as evaluation parameters. Finally, this paper analyzes the actuator motion characteristics by Some experiments.

For this article, I personally have the following questions:

1: The statement of distortion phenomena of the half engagement point of slow disengaging solenoid valve is lacking in this paper.

2: You should provide information about the experiments data that you are using, and what parameters are used during experiments.

  1. The influence of the road conditions are not being well considered, which I think it is necessary.

Note: There are two descriptive errors and typographical errors in the article:

(1) In line 250, the error is described as "Table 1shows that the actuator "(should be Table 1 shows that the actuator)

(2) In line 285, the error is described as "the rang of the engaging velocity range". (should be the range of the engaging velocity)

(3) Pictures should be aligned to the center of the page.

Round 2

Reviewer 1 Report

I am still concerned about the contribution. What remains unclear to me is whether the overall Section 2 is a fundamental part of the paper or not. I do not understand if experimental results validate the model in Section 2 or, instead, they are only interpreted according to the model. In this latter case I do not understand why deriving a quite detailed model rather than a simpler one. The paper should clarify such aspect. For instance, having the model in Section 2 I would have expected some numerical simulations that could be compared with experiments of section 4.  

One further suggestion: please verify the equations numbering since they are completely misplace. For instance at line 102 the equation (5) is referenced but I guess you refer to equation (2). At line 108 and 116 equation (6)  is referenced but I guess it is eq. (3), instead. Such issues are present in several other points of the paper. 
